# An Italian Case Study for Assessing Nutrient Intake through Nutrition-Related Mobile Apps

**DOI:** 10.3390/nu13093073

**Published:** 2021-08-31

**Authors:** Lorenza Mistura, Francisco Javier Comendador Azcarraga, Laura D’Addezio, Deborah Martone, Aida Turrini

**Affiliations:** 1CREA Council for Agricultural Research and Economics—Research Centre for Food and Nutrition, Via Ardeatina 546, 00178 Rome, Italy; fjavier.comendador@crea.gov.it (F.J.C.A.); laura.daddezio@crea.gov.it (L.D.); deborah.martone@crea.gov.it (D.M.); 2Independent Expert (Formerly Council for Agricultural Research and Economics), 00178 Rome, Italy; aida.turrini@gmail.com

**Keywords:** diet tracker apps, 24 hours dietary recall, dietary monitoring, food consumption survey

## Abstract

National food consumption surveys are crucial for monitoring the nutritional status of individuals, defining nutrition policies, estimating dietary exposure, and assessing the environmental impact of the diet. The methods for conducting them are time and resource-consuming, so they are usually carried out after extended periods of time, which does not allow for timely monitoring of any changes in the population’s dietary patterns. This study aims to compare the results of nutrition-related mobile apps that are most popular in Italy, with data obtained with the dietary software Foodsoft 1.0, which was recently used in the Italian national dietary survey IV SCAI. The apps considered in this study were selected according to criteria, such as popularity (downloads > 10,000); Italian language; input characteristics (daily dietary recording ability); output features (calculation of energy and macronutrients associated with consumption), etc. 415 apps in Google Play and 226 in the iTunes Store were examined, then the following five apps were selected: YAZIO, Lifesum, Oreegano, Macro and Fitatu. Twenty 24-hour recalls were extracted from the IV SCAI database and inputted into the apps. Energy and macronutrient intake data were compared with Foodsoft 1.0 output. Good agreement was found between the selected apps and Foodsoft 1.0 (high correlation index), and no significant differences were found in the mean values of energy and macronutrients, except for fat intakes. In conclusion, the selected apps could be a suitable tool for assessing dietary intake.

## 1. Introduction

In recent years, a wide range of health-related apps has been developed for smartphones and tablets in order to help users monitor their body weight, diet, physical activity and wellness as a whole. In 2017, the number of such health-related apps was 325,000 [1] and the number of downloads was 3.7 billion worldwide [2].

As part of this huge number of apps, those that monitor the diet have sparked interest amongst nutritionists because it allows for the monitoring of the user’s nutritional status by recording the data of foods and beverages consumed, as well as their physical activity level and body weight. Generally, the main purpose of these apps is to help individuals control their body weight by suggesting healthy dietary behavior [3].

In this context, the scientific literature has offered a large production of studies concerning these digital tools which aim to promote healthy dietary behaviors [4,5], both among people who want to actively manage the maintenance of their health condition [6,7], and among those suffering from food allergies, intolerances, or other nutrition-related diseases [8].

At the same time, another segment of the literature has focused on analyzing the input and output features of the apps and comparing them with the standard method to assess the food intake [5,9]. 

From the food and nutrition science perspective, the gathering of dietary data through a national survey using the food dietary record, or 24 h recall is the main approach for defining food consumption patterns in population groups, identifying the main sources of nutrients and assessing the adequacy of diets.

Moreover, these data represent the backbone of drafting the national nutritional guidelines, the achievement of exposure assessment studies and the development of more sustainable dietary models in terms of an environmental footprint.

In 2009 at the European level, the European Food Safety Authority (EFSA) underlined the importance of these data launching the EU Menu project to make the collection of more harmonized food consumption data among the EU Member States, to be used in dietary exposure assessments of food-borne hazards and nutrient intake estimations [10].

In Italy, national dietary surveys are conducted about every 10 years, and the fourth has just been completed and was carried out following the above EU Menu methodology suggested by the EFSA guidelines [11].

The first Italian survey on food consumption dates back to 1980–1984, subsequently two other surveys were carried out respectively in 1994–1996 and 2005–2006, and the fourth national dietary data collection was the recently completed IV SCAI study [12]. All four surveys used different tools and methods depending on both the type of information that was considered important at the time of the survey, the availability of effective and adequate tools for data collection and economic feasibility.

Nowadays technological innovations, such as the development of software to input the 24 h recall data or dietary records helps considerably in the data management and analyses of food consumed. In any case, the most critical point in this type of survey is the recruitment of volunteers who must devote time (more or less two hours per each day of the survey, depending on the method adopted), to fill in the dietary record or answering a list of questions asked by trained staff [13].

Nevertheless, people’s interest in monitoring their food consumption through digital solutions is also growing in Italy, where in 2020 diet-related apps were downloaded approximately 79 thousand times occupying the second position in the Google play store’s ‘health and fitness’ category ranking [14].

The amount of big data produced by these apps could be used to integrate in real-time the information derived from the traditional food consumption surveys that are usually time and cost-consuming [15]. It is a matter of fact that food consumption surveys are conducted to take a picture of the population’s food and beverage intake over a given time period and do not capture any changes in food habits that may occur in a short-term period due, for example, to a sudden adverse event, such as economic shocks or the current pandemic crisis. This is the reason why a large set of surveys on changes during the lockdown due to the COVID-19 pandemic have been carried out all over the world [16,17,18] and in Italy [19,20,21,22].

The aims of this work are to analyze the main features of the most popular diet-related mobile apps in Italy; to evaluate their potential ability to estimate nutrient intake based on a 24 hours dietary recall; and finally, to compare the apps’ outputs with that obtained from the web-based validated software Foodsoft 1.0 used in the aforementioned survey-IV SCAI, carried out according to the EFSA EU Menu methodology [11].

## 2. Materials and Methods

### 2.1. App Selection

The apps considered in the study were selected from the Google Play Store for Android and the iTunes Store for iOS in February 2020 from a personal computer without logging in from any account to avoid any influence in the search results. The filters were: The Italian language, free of charge and European developer. 

In Google Play, seven keywords were used to search for the apps, entered one by one, for the selection: “diet”, “weight loss diet”, “food diary”, “count calories”, “weight loss”, “nutritionist” and “24 h recall”. For each keyword, Google Play offered a maximum of 250 apps.

In iTunes Store, all the apps of the category “health and well-being” were considered because there was not the possibility to refine the research by keywords. 

Ten typologies have been defined to categorize the apps: Diet tracker, Calories tracker, Diet, Health tracker, Physical activity, Recipes, Water monitoring, Weight loss, Well-being and Non-diet related.

The main criteria adopted to classify the apps were: the app name, when it was sufficiently explicative of the main features (e.g., sculpted abs in the category Physical activity); the description given by the Google Play or iTunes stores; the reading of the user reviews and the information on the app web site.

In the case that the app offered more than one feature (e.g., “30-day weight loss” which in addition to the exercise program also provided a diet), the allocation was based on the main features always suggested by the name of the app.

The Diet tracker apps have been tested to verify if they would be actually able to record a food diary.

### 2.2. Nutritional Data 

Twenty 24 hours dietary recall (24-HRs) data have been randomly extracted from the IV SCAI study food consumption database. 

The selection criteria of the twenty 24-HRs were based on: the age of the participant ranges between 30–50 years old, normal weight (Body Mass Index ranges between 18.5–24.9) and energy intake coverage higher than 60%.

In the IV SCAI food consumption survey the interviews were conducted according to the Multiple Pass Food Recall five-step approach to obtain information on consumed food, portion size, preparation method, the brand of the food/beverage/supplement, time and place of the meal consumed. The food intake quantification was based on a validated food/dishes picture book. Composite dishes were broken down into their ingredients based on standard recipes. In addition, the quantification of portion sizes was also accomplished with household measurements and models of tableware (cups, glasses, spoons, plates, and bowls). 

Foodsoft 1.0 was the management software, previously validated [23], used to enter data during the 24-HRs interviews. Foodsoft 1.0 includes four databases: (i) ‘Food descriptors’ (ii) ‘Household unit of measurement’; (iii) ‘Standard recipes’; and (iv) ‘Food composition’ that are used to quantify food and nutrient intake.

Foodsoft 1.0 was used as a research reference method to compare the output of the five selected apps. 

Procedure:

All the data extracted from the food consumption database were entered by the authors in each of the five apps (each author entered five 24-HRs) following the standardized procedure below described.

In the first step, a perfect knowledge of food weights and ingredients of composite dishes was envisaged. Food items were selected in the app food list that matched the description reported in the 24-HRs regardless of brand specification. In the case where a food item was not present in the app list of foods, the most similar food in the description and calories was chosen (e.g., generic fruit yogurt instead of peach yogurt). It was also decided to input the amount in grams of food and beverage items corresponding to the portion/household measures selected in Foodsoft 1.0 instead of the portion sizes proposed by the apps.

The recipes were disaggregated into individual ingredients. Their formulation and relative weights have been entered using Foodsoft 1.0 as a reference because most of the apps do not show the ingredients list and the corresponding amount.

A second step was performed using seven out of the twenty 24-HRs recalls selected among those showing energy intakes of the same magnitude comparing the energy intake assessed by Foodsoft 1.0 and the corresponding values obtained through the apps. This second approach was based on the hypothesis that whoever entering the data was a ‘consumer’ with poor knowledge of the technique of reporting individual food consumption. In this case, the portion sizes and the household measures proposed by the apps equaled the ones described in the 24-HRs which were used to quantify the amounts of foods and beverages consumed. If the app’s unit of measure was different from the corresponding described in the 24-HRs, a conversion was performed referring to the DRV of Nutrients and Energy for the Italian population [24] or using specific websites. In this step, the recipes were not disaggregated into ingredients and in case they were missing in the list proposed by the app, the most similar items were selected based on the narrative food description.

### 2.3. Statistical Analysis

Energy and nutrient intakes of the twenty 24-HRs estimated from the five apps and Foodsoft 1.0 were described using medians, means and standard error of means (SE). The paired samples t-test was computed to compare the mean intakes of energy, carbohydrates, protein, and fats estimated by Foodsoft 1.0 and those estimated by each of the five diet-tracking apps. The Spearman rank correlation coefficient (r) was also calculated to study the relationship between the reference data and those from the applications. A *p*-value < 0.05 was considered statistically significant.

Lastly, the Bland–Altman method [25] was used to plot the agreement between the means of energy and macronutrients from the two dietary assessment tools: Foodsoft 1.0 and each of the five apps. The statistical analyses were performed using R version 4.0.2.

## 3. Results

### 3.1. Selection and Description of the Apps

In Google Play, out of 1750 apps examined, 526 were excluded because of duplicates and 809 were not in Italian; in the iTunes Store, 240 apps were selected and 14 rejected since they were not in Italian (Figure 1).

The remaining apps 415 and 226, respectively in the Google Play and iTunes Store, were classified into ten categories (Table 1). The most represented app category was Physical Activity for both stores (*n* = 104 and *n* = 97) followed by Weight Loss for Google Play and Not related for iTunes. The Diet Tracker category included 14 apps in Google Play and seven in iTunes that have the feature of allowing for the registration of a food diary. YAZIO, Lifesum, Oreegano, Macro and Fitatu are the five apps selected, the others have been excluded because they had already been reviewed in the literature (MyFitnessPal, FatSecret, Noom Coach, and Lose It!) [5,9] or the number of downloads was less than 50,000 (Kcalories)or were not developed in a European country (Food Dairy). 

The main input data of all apps (free access) was age, weight and height, gender, physical activities level. Moreover, the user has to set their own goal in terms of weight or gain loss or diet monitoring. In Oreegano and Fitatu is also possible to select the type of diet (vegan, pescatarian and vegetarian) and food intolerance (lactose, gluten, fish). The framework to fill in the data was very similar to the classic dietary diary divided into three main meals (breakfast, lunch, dinner) and snacks for all apps. The food item search text was available in all apps, as well as a section to add a personalized recipe indicating the portion size and the amount of the single ingredients. It was also possible to update the food list with a new commercial food product using the barcode scanner (except for Oreegano) that identified the item, prompting the user to fill in the nutritional facts written in the label.

Compared to the dietary record, the main missing information is the place where the food is consumed and the meals schedule, except for Fitatu where is possible to record the time of consumption. All five apps have the text search function for food names and a section to select the recipes and adding the personalized recipes indicating the amount of the main ingredient (Table 2).

As output, the energy and macronutrient (excluded alcohol) intake are calculated based on the goal that the user intends to reach.

All the selected apps have the text search for selecting the food items with the number of calories for 100 g of food or serving.

### 3.2. Nutritional Analysis

Table 3 shows the results of the first step in terms of mean, standard error of the mean (SE) and median for energy, carbohydrates, protein and fat intakes estimated by Foodsoft 1.0 and the five-digital tools in the twenty 24-HRs data. No statistically significant differences in energy, carbohydrates and protein mean intakes between the reference method and all applications reviewed were observed. Conversely, Macros and Oreegano showed a significantly lower value of mean fat intake with respect to Foodsoft 1.0. 

The Spearman correlation coefficient was high both for energy and macronutrients intakes. It ranged between 0.98–0.95 (Lifesum–Oreegano) for energy intake; 0.96–0.93 (Oreegano–Fitatu) for carbohydrates; 0.96–0.89 (Oreegano, Fitatu–YAZIO) for proteins and 0.96–0.84 (Fitatu–Lifesum) for fats.

The results of the second step adopted for registering the seven 24-HRs selected, presented a significant difference in the mean protein intake for Fitatu and Macros compared to Foodsoft 1.0. A not significant difference in mean intake between the five digital tools and the software was observed for the remaining macronutrients, energy and grams (Table 4). 

In the first step, the strength of association between the energy intake from Foodsoft and that of each of the five apps varies between 0.93 and 0.79 (Fitatu, Oreegano) (Figure 2a). In the second step, for carbohydrates YAZIO presents a very strong correlation (0.96) and Lifesum is uncorrelated (0.04); the association between protein ranges between (0.72–0.27) (YAZIO -Fitatu) and lastly, for fat, Oreegano presents a correlation of 0.29 whilst all the other apps are uncorrelated compared with Foodsoft 1.0 although a good correlation is observed between them (Figure 2b). 

The analysis of agreements for the whole sample shows that 5% of cases fell outside of the limits of agreement for estimates of energy intake using Lifesum, Oreegano, Macros and Fitatu compared with Foodsoft 1.0, and for YAZIO no cases are outside the interval. In the analysis of the seven 24-HRs, one case is outside the interval for YAZIO and for the other apps all the cases are within the range (Figure 3).

For the assessment of carbohydrates, the plots indicate a good agreement for Oreegano, Macro, Fitatu and YAZIO with 5% of cases outside the limit of interval and 10% of cases for Lifesum. In the selected sample, Oreegano presented only one 24-HRs outside the interval (Figure 4). 

The estimate of protein in all the apps shows that 5% of the cases are outside of the interval, Lifesum and Oreegano have one case respectively outside the interval for fats. No cases are outside the range when the analysis considered the seven 24-HRs (Figure 5 and Figure 6).

## 4. Discussion

The present study describes the main features of the most popular apps in Italy for monitoring the food consumed by the user and compares the app’s nutritional data with those of a dietary software which, for the purposes of this work, is the Foodsoft 1.0 for data entry and management. It was developed and used in the last Italian national dietary survey (IV SCAI). 

In the first step, the results show that there is a good agreement between the five selected digital tools and Foodsoft 1.0, more precisely the correlation index was high, and no significant differences of means were found for energy, carbohydrate and protein except for fat intakes. The Bland-Altman plots show that the apps with the highest bias are Oreegano (35.6) for energy estimation, YAZIO (7.5) for carbohydrates, Lifesum (−3.5) for protein and Oreegano (8.4) and Macros (5.4) for fats. Furthermore, the mean difference suggests that Oreegano is the most variable (critical difference = 235) for energy estimation, Fitatu for carbohydrates (41.4), YAZIO for proteins (25.6) and finally Lifesum for fats (27.9). The variability around the mean difference is mainly attributable to the selection of the right food items by the experts.

The difference in the mean fat intakes could be due to food composition databases used in the apps because there is variability in the food chemical composition both between apps and Foodsoft 1.0. For example, in Macros, the fat amounts reported for ‘ham cooked’ is 2.5 g for 100 g and 14.7 g in the Foodsoft 1.0 databases and in general fats the amount is lower for the pork meat (4 g of fats for 100 g against 10 g in Foodsosft). In Oreegano, the lower fats mean intake can be imputed to the olive oil that is 90 g for 100 g of product compared to 99.9 g of fats used for the IV SCAI survey, considering that olive oil is a frequently consumed food both as an ingredient of recipes and as a seasoning, and is present in many eating occasions.

In the apps, there is no information on which food composition datasets are loaded but in the first step, the Spearman’s correlation plots show that the datasets are very similar since the correlation is high between the apps and not just with Foodsoft 1.0. There are some dissimilarities at the level of the food item, but they do not affect the overall results. As an example, the energy of the ‘Buffalo Mozzarella cheese’ is 209 kcal per 100 g in Macros and 288 kcal per 100 g in the database of the Foodsoft 1.0. The energy of ‘eggplant row’ is 24 kcal/100 g in Lifesum compared to 18 kcal/100g in Foodsoft 1.0. 

In the second step, seven 24-HRs were selected whose nutritional values from the apps were closest to the values of Foodsoft 1.0 and were entered in the apps by the same experts identifying with ordinary consumers who have no familiarity with the consumption data in all their aspects. In this second step the main outcome confirmed no difference in the mean energy and macronutrients intake between the apps and Foodsoft 1.0, except for proteins. On the other hand, a careful examination of the plots suggests that the bias is high in Fitatu and Macros for energy and protein (148; 13.1 and 159.7; 9.7 respectively), for carbohydrates in Macros (44.9) and for fats in Oreegano (16.3). The mean difference intervals are wide, reflecting the small sample size and the great variation of the differences. This can also be attributed to the difference in the number of grams because the portions provided by the apps were used in place of the grams of the standard portions suggested by Foodsoft 1.0 used in the first step. In addition, most recipes are entered as composite foods and not disaggregated into ingredients as in the first approach, so ingredient variability resulted in a worse correlation with Foodsoft 1.0 values of macronutrients than among apps. This confirms the importance of knowing in sufficient detail what and how much is eaten to obtain good estimates of dietary intake.

The portion size provided by the apps and the possibility to choose a food in the food list uploaded, especially recipes, are other aspects to keep under control. For example, the ‘pasta alla carbonara’, a typical Italian dish, in Macros 100 g corresponds to 379 kcal and a ‘Plate’ (generic, no indication of the gram) to 550 kcal, in Lifesum the standard portion is set at 100 g and the energy at 378 kcal, in Fitatu a generic portion, without specifying the amount, is equivalent to 241 kcal and there is also the option of 100 g of frozen pasta equal to 172 kcal, in Oreegano portions are not available and 100 g of Carbonara are equivalent to 760 kcal; finally, YAZIO which does not provide for the indication of portions, 100 g corresponds to 379 kcal or 490 kcal. In most apps, the kcal refers to the quantity of a recipe and not to the amount of raw pasta for that recipe which is the food item whose quantity a consumer usually knows; therefore, for those who have no experience in managing nutritional data it is difficult to choose the correct quantity and type of the recipe unless it is broken down into the main ingredients. Another example is “Pizza Margherita”, in Macro 100 g corresponds to 148 kcal and a Pizza Margherita (generic, without indication of the gram) to 580 kcal, in Lifesum the standard portion is set at 100 g and the energy at 250 kcal, in Fitatu 100 g is equivalent to 271 kcal, In Oreegano 100 g of raw Pizza Margherita weight is equivalent to 477 kcal, 100 g of cooked weight to 181 kcal, finally YAZIO that does not provide portions, 100 g correspond to 266 kcal.

These findings suggest that the digital tools could be feasible for assessing dietary intake limited of energy and the main macronutrients. The outcomes of this study are very close to those of other similar studies comparing apps or web-based 24-HRs [26]. For example, Faillaize et al and Ferrara et al. [5,9] described that the apps selected provided estimates of energy and saturated fat intake comparable to the UK research standard method Dietplan6 version 6.0 and USDA Food Composition Database, respectively.

Furthermore, the accuracy and completeness of the dietary assessment is always challenging but when considering self-administered recording the risk of imprecision is higher. Web-based applications usually guide the user, but apps do not show such functionalities in the free version; also, the self-administrated 24-HRs are subject to imprecision if the users are not well trained in advance through guided procedures or video tutorials [27].

Reliability and completeness of food composition databases is another crucial aspect, considering that the development and maintenance of food composition tables are already challenging themselves [28]. Therefore, coordinating efforts in creating shared international food composition databases, such as that of the EuroFIR AISBL is [29] worth to be sustained.

Also, Vasiloglou et al. [30] in their multinational survey on the use of nutritional diet app identified as barriers, the use of inaccurate databases for food composition 52% of respondents, the fact that local food composition is not supported 48.2% and that the user needs to possess technical expertise 43.3%. 

The limitation of this study is the low number of the 24-HRs selected for analysis, although they proved to be sufficient to highlight the main differences between Foodsoft 1.0 and the selected apps, both in results and in functionalities. Another limitation is that the data of the 24-HRs, both in Foodsoft 1.0 and in the mobile apps, were entered by professionals specialized in dietary data management well trained in entering the consumption data recorded through the 24-HRs with computer support, and performing the quality control of the data collected and entered [12], paying attention also to the energy and macronutrients values of the food items reported in the app. 

Moreover, the comparison was done using the app’s free version that is limited in terms of macronutrient and micronutrients values, availability of more recipes and the possibility to download the imputed data of intake. 

## 5. Conclusions

The diet-tracker apps examined could be considered suitable for collecting information on consumer nutritional data, although support tools (such as pictures of portion sizes, the ingredient list of the recipes) are necessary to guide the user to an appropriate selection of the food items and for a more correct evaluation of the quantities of food consumed.

A collaboration between app developers and nutritionists, experts in nutrition data management, and public health administrators could be wise to improve the quality of the app. Indeed, nutrition-related apps are tools to collect information that could also be used in public canteens and schools to monitor eating behaviors. Furthermore, the consumer’s point of view is to be considered, both because the use of nutrition-related apps can help increase their knowledge of the nutritional values of the food consumed and by the developer who can improve the app’s features [31].

In the future, thanks to the rapid advance of the Internet of Things [32] including Artificial intelligence and wearable tools [33], a large number of novel applications, such as smart dining tables that automatically track what and how much each individual eats over the course of a meal [34] and smart cups [35] to detect the amount and typology of the liquid inside, may encourage the collection of data on food consumption and develop appropriate validation analyses.

## Figures and Tables

**Figure 1 nutrients-13-03073-f001:**
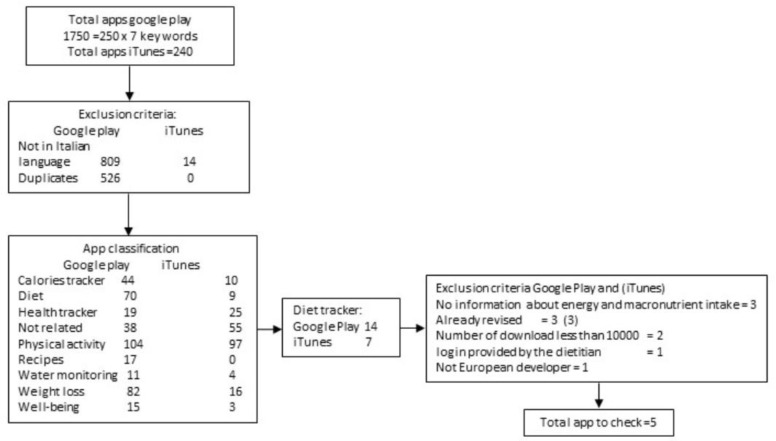
Flowchart of apps research and selection.

**Figure 2 nutrients-13-03073-f002:**
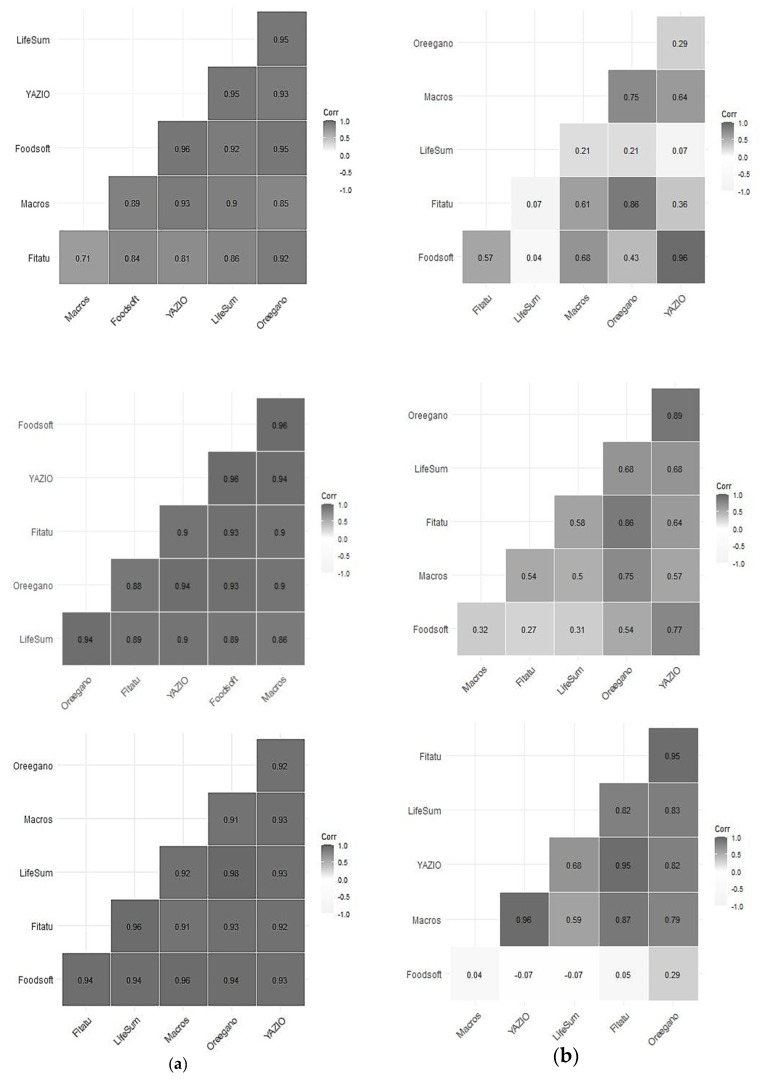
Spearman correlation plots for estimated intakes of carbohydrates, proteins and fats for the whole sample (**a**) and for the selected seven 24-HRs (**b**).

**Figure 3 nutrients-13-03073-f003:**
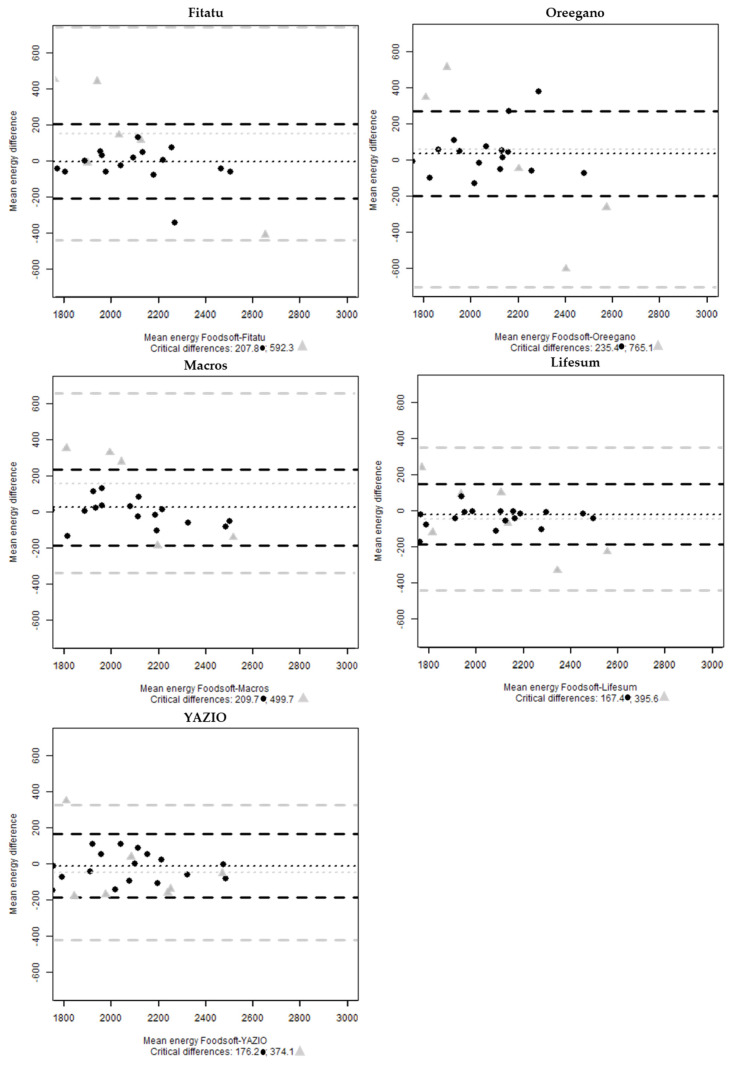
Bland-Altman plots of the mean energy (kcal) difference between Foodsoft 1.0 and Fitatu, Oreegano, Macros, YAZIO and Lifesum whole sample (●) and the selected 24-HRs (

).

**Figure 4 nutrients-13-03073-f004:**
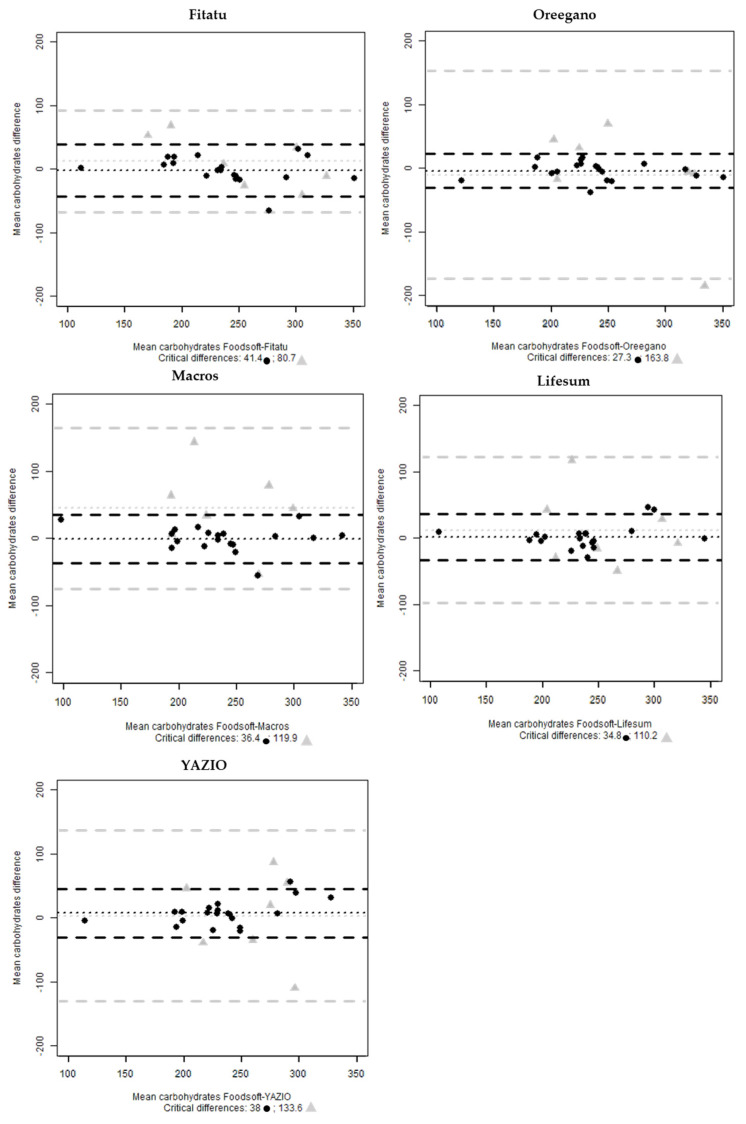
Bland-Altman plots of mean carbohydrates (g) difference between Foodsoft 1.0 and Fitatu, Oreegano, Macros, YAZIO and Lifesum whole sample (●) and the selected 24-HRs (

).

**Figure 5 nutrients-13-03073-f005:**
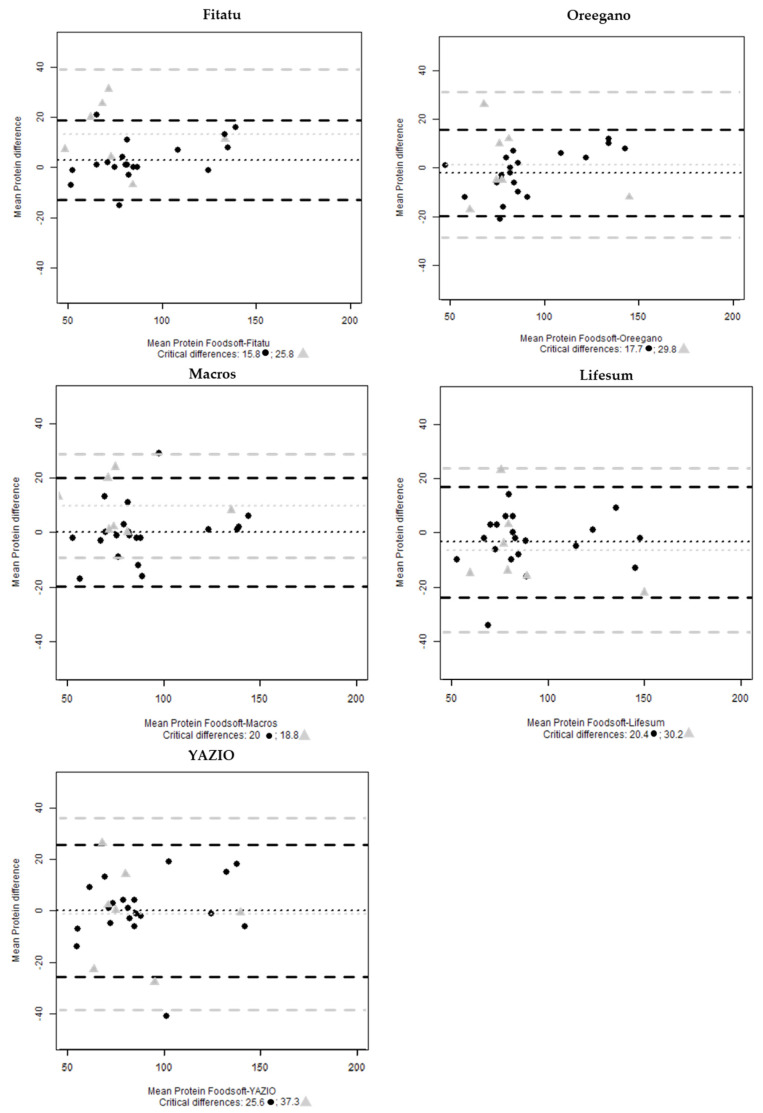
Bland-Altman plots of mean protein (g) difference between Foodsoft 1.0 and Fitatu, Oreegano, Macros, YAZIO and Lifesum whole sample (●) and the selected 24-HRs (

).

**Figure 6 nutrients-13-03073-f006:**
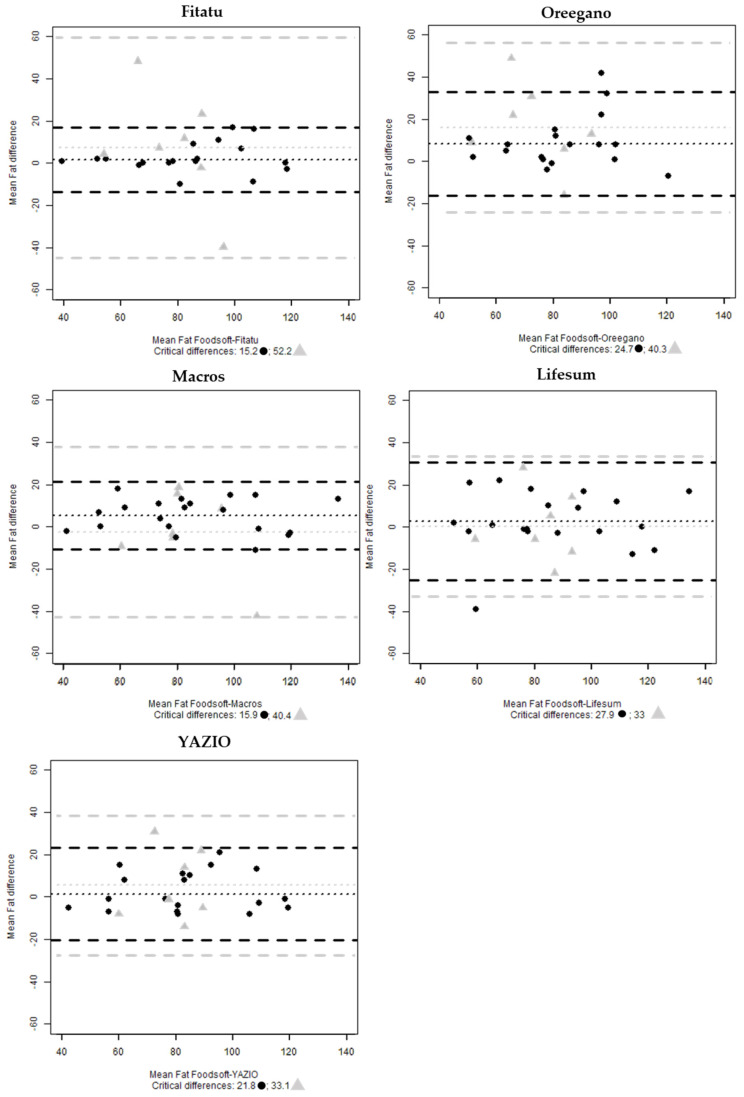
Bland-Altman plots of mean fats (g) difference between Foodsoft 1.0 and Fitatu, Oreegano, Macros, YAZIO and Lifesum whole sample (●) and the selected 24-HRs (

).

**Table 1 nutrients-13-03073-t001:** Description of app categories.

App Categories	Description of the Main Features
Diet tracker	Allow the user to record the foods and drinks consumed during daily meals. They are the object of evaluation of this work.
Calories tracker	Display the calories of food items.
Diet	Offer specific diet plans, such as Zone Diet, Ketogenic Diet, Dukan Diet, Blood Group Diet, Intermittent Fasting, etc.) or develop personalized diet
Health tracker	Include calculation tools useful for managing conditions, such as diabetes, weight control, blood pressure, blood glucose, etc. and usually require the supervision of a medical doctor or dietician
Physical activity	Used for specific physical activity programs, such as bicycle, running, walking, training and so on. fitness: the state of physical well-being or physical form of the individual
Recipes	Provide a collection of recipes for a healthy diet or vegetarians, vegans, etc.
Water monitoring	Track the water drank and hydration status.
Weight loss	Designed for weight loss programs and/or fat reduction of specific body regions, through specific exercises (fat burning workout and fitness exercises)
Well-being	Teach users about meditation techniques to improve sleep and the ability to relax better.
Non-diet-related	Includes several apps regarding management of patients, calendar of menstrual cycle, personal logbook

**Table 2 nutrients-13-03073-t002:** Main features of the Apps.

Main Features	YAZIO	Lifesum	Macros	Fitatu	Oreegano
Text search	x	x	x	x	x
Barcode scanner	x	x	x	x	
Serving size	x	x	x	x	x
Meals	x	x	x	x	x
Adding a new food/recipes	x	x	x	x	x
Energy and macronutrient at food items level				x	x
Data export				x	

**Table 3 nutrients-13-03073-t003:** Mean, standard error of mean (SE) and median of energy and nutrient intake estimated by Foodsoft 1.0 and the five apps for the whole 24-HRs.

	Foodsoft 1.0		Fitatu			LifeSum			Macros			Oreegano			YAZIO		
Main output	mean (SE)	median	mean (SE)	median	*p* * values	mean (SE)	median	*p* * values	mean (SE)	median	*p* * values	mean (SE)	median	*p* * values	mean (SE)	median	*p* * values
Energy	2096.9 (83.8)	2063.5	2098.6 (81.3)	2052.0	0.943	2115.7 (73.3)	2122.5	0.332	2071.0 (77.2)	2005.5	0.292	2061.4 (86.9)	2036.5	0.201	2108.6 (80.2)	2081.0	0.567
Carbohydrate	237.6 (11.5)	237.5	239.7 (12.6)	242.5	0.654	236.0 (10.5)	239.0	0.693	238.3 (12.3)	235.5	0.867	241.6 (11.8)	239.0	0.214	230.3 (9.3)	230.5	0.111
Protein	89.4 (6.3)	81.5	86.4 (5.6)	81.0	0.118	92.8 (6.1)	85.0	0.155	89.3 (5.8)	82.5	0.982	91.3 (5.1)	85.5	0.346	89.3 (5.8)	83.5	0.986
Fat	88.3 (5.7)	87.5	86.7 (6.0)	86.0	0.368	85.6 (5.5)	79.5	0.398	83.0 (5.9)	78.5	<0.05	80.0 (6.1)	78.0	<0.05	87.1 (6.5)	83.5	0.621

* Paired t test was used for mean comparison between apps and Foodsoft 1.0.

**Table 4 nutrients-13-03073-t004:** Median, mean and standard error of mean (SE) of nutrient intake estimated by Foodsoft 1.0 and the five apps for the seven 24-HRs selected.

	Foodsoft 1.0		Fitatu			LifeSum			Macros			Oreegano			YAZIO		
Main output	mean (SE)	median	mean (SE)	median	*p ** values	mean(SE)	median	*p ** values	mean (SE)	median	*p* * values	mean (SE)	median	*p* * values	mean (SE)	median	*p* * values
Grams	1419.7 (169.3)	1187.0	1461.7 (198.3)	1454	0.812	1610.4 (227.8)	1501	0.812	1311.6 (186.6)	1091	0.375	1508.4 (185.8)	1317	0.468	1528.6 (191.9)	1398	0.578
Energy	2074.1 (84.4)	2103	1926.1 (177.9)	1904	0.218	2115.6 (137.9)	2055	0.687	1914.4 (149.6)	1830	0.156	2013.0 (197.8)	1640	0.687	2118.7 (109.1)	2065	0.375
Carbohydrate	261.1 (17.9)	242	249 (28.5)	267.7	0.578	249.2 (22.4)	257	0.937	216.3 (21.7)	207	0.109	270.1 (34.2)	214	0.937	257.7 (19.7)	262	0.812
Protein	83.9 (10.1)	81	70.7 (11)	55.7	<0.05	90.3 (12.5)	79	0.375	74.1 (10.7)	71	<0.05	82.6 (11.8)	75	1.000	85.3 (11)	75	1.000
Fat	82 (5.3)	87	74.8 (9.2)	76.5	0.296	81.9 (5.7)	83	0.937	84.1 (8.1)	80	1.00	65.7 (7.8)	57	0.109	76.4 (4.8)	78	0.611

* Paired Wilcox test was used for the mean comparison between apps and Foodsoft 1.0.

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
