# Peer review of "An Italian Case Study for Assessing Nutrient Intake through Nutrition-Related Mobile Apps"

_nutrients, 2021, doi:10.3390/nu13093073_

Round 1

Reviewer 1 Report

Overall, I found this manuscript to be an interesting read.

Throughout the text there are a number of places where clarity could be improved and these changes, along with more specific comments, are noted below:

  • First of all, this is a national, italian study, maybe the title should reflect this and in the abstract shall be mentioned, as it is in the discussion section.
  • The conclusion section is missing!
  • In the discussion several limitations are highlighted but missing a theoretical recommandation
  • References have double nummbering, please check.
  • Suggestions for some additional references, up-to-date literature?!
  • Figures are blurred, the quality shall be improved!

Reviewer 2 Report

The submitted manuscript is interesting and easy to read. The introduction provides sufficient background to the topic. Material and methods section is quite accurate. Authors should more clarify why those exclusion criteria were used specifically. Results section in terms of discussion is scarce at some points and should be improved. Article would also benefit from separation of conclusion section. References should be edited to MDPI requirements.

Detailed comments:

Line 2 I suggest total overhaul of the tile, currently it is simple and overcomplicated at the same time. Either stick to something simple or specific.

Line 183 please specify which apps meet the selected requirements and were excluded because of being already reviewed in the literature

Line 186 – table 1 is to overcrowded with information, I suggest to place just a list specific functions of apps

Line 188 – In 1750 google play apps are there any duplicates? If yes whole text should be corrected for amount of google play apps with regards to duplicates

Line 224 please use rather software name than gold standard

Line 239 Graph should be re-designed (unreadable) maybe 2 charts in a row would be better, moreover graph is a printscreen and includes autocorrect redline underlining of titles

Figure 3-6 the discussion of provided figures I scarce, please expand.

Line 352-360 I suggest separating this part to conclusion sections

Round 2

Reviewer 2 Report

The authors have improved the manuscript substantially. In its current form it is interesting, and easy to read and understand. Therefore I can be recommended for publication in Nutrients.